# Downregulation of Bmal1 Expression in Celiac Ganglia Protects against Hepatic Ischemia-Reperfusion Injury

**DOI:** 10.3390/biom13040713

**Published:** 2023-04-21

**Authors:** Jiarui Feng, Lilong Zhang, Enfu Xue, Zhendong Qiu, Ning Hu, Kunpeng Wang, Yingru Su, Weixing Wang

**Affiliations:** 1Department of General Surgery, Renmin Hospital of Wuhan University, Wuhan 430060, China; 2Key Laboratory of Hubei Province for Digestive System Disease, Wuhan 430060, China

**Keywords:** hepatic ischemia-reperfusion injury, celiac ganglion, Bmal1

## Abstract

Hepatic ischemia-reperfusion injury (HIRI) significantly contributes to liver dysfunction following liver transplantation and hepatectomy. However, the role of the celiac ganglion (CG) in HIRI remains unclear. Adeno-associated virus was used to silence Bmal1 expression in the CG of twelve beagles that were randomly assigned to the Bmal1 knockdown group (KO-Bmal1) and the control group. After four weeks, a canine HIRI model was established, and CG, liver tissue, and serum samples were collected for analysis. The virus significantly downregulated Bmal1 expression in the CG. Immunofluorescence staining confirmed a lower proportion of c-fos^+^ and NGF^+^ neurons in TH^+^ cells in the KO-Bmal1 group than in the control group. The KO-Bmal1 group exhibited lower Suzuki scores and serum ALT and AST levels than the control group. Bmal1 knockdown significantly reduced liver fat reserve, hepatocyte apoptosis, and liver fibrosis, and it increased liver glycogen accumulation. We also observed that Bmal1 downregulation inhibited the hepatic neurotransmitter norepinephrine, neuropeptide Y levels, and sympathetic nerve activity in HIRI. Finally, we confirmed that decreased Bmal1 expression in CG reduces TNF-α, IL-1β, and MDA levels and increases GSH levels in the liver. The downregulation of Bmal1 expression in CG suppresses neural activity and improves hepatocyte injury in the beagle model after HIRI.

## 1. Introduction

Ischemia can severely impede blood circulation, which is necessary for normal organ functioning [1]. Ischemia-reperfusion injury (IRI) is a phenomenon where organ damage due to hypoxia is paradoxically exacerbated after the restoration of blood circulation [2]. As the largest solid organ, the liver is particularly susceptible to hypoxia because it relies heavily on oxygen for energy metabolism. Thus, hepatic ischemia-reperfusion injury (HIRI) is a common and severe complication arising from various clinical conditions, including liver transplantation, surgery, severe trauma, and hemorrhagic shock [3]. HIRI can be divided into two phases: ischemia and reperfusion. The process of HIRI involves multiple mechanisms, including redox imbalance, calcium overload, and the activation of inflammatory cells [4,5,6,7]. During the ischemic phase, the ATP concentration in cells decreases, resulting in the failure of the Na/K pump, cellular edoema, and an increase in cytoplasmic calcium concentration, ultimately leading to cellular damage [6,7,8]. During the early reperfusion phase, which occurs within 2 h of reperfusion, proinflammatory mediators (such as TNF-α, IL-6, IL-1, and arachidonic acid) and reactive oxygen species (ROS) are released by Kupffer cells. In the late reperfusion phase, which occurs between 6 and 48 h after reperfusion, neutrophil-mediated inflammatory reactions occur. Complement factors, chemokines, and cytokines recruit neutrophils into the liver and damage cells by releasing ROS or proteases [5,6,7,8].

Several protective measures have been reported to alleviate HIRI, including pharmacological interventions, ischemic preconditioning (IPC), ischemic post-conditioning (IPostC), and mechanical reperfusion [9,10,11,12]. Various pharmacological agents have been employed to thwart HIRI, primarily to counteract the increased oxidative stress and facilitate immunomodulation, such as methylprednisolone, trimetazidine, and ulinastatin [9]. Notably, variations in existing liver clamp techniques designed to reduce HIRI by manufacturing IPC and IPostC have been extensively studied and are routinely implemented in liver transplantation and resection [1]. However, most of these strategies are still in the preclinical animal model phase, and the precise therapeutic targets and underlying mechanisms remain unclear.

Circadian rhythms are nearly 24-h endogenous oscillations of biological processes in organisms that are associated with the earth’s rotation cycle [13]. A vast majority of organs in the body, including the liver, exhibit a distinct circadian rhythm [14]. This rhythm plays a crucial role in liver homeostasis, including hepatic metabolism. Biological clock dysfunction can accelerate the development of liver diseases, such as fatty liver, hepatitis, cirrhosis, and liver cancer, which, in turn, can also disrupt the biological clock function [15]. The hepatic autonomic nervous system transmits information about hepatic osmolarity, glucose, and lipids from the liver to the central nervous system. It also regulates hepatic blood flow, metabolism, and bile secretion based on efferent signals from the central nervous system [16]. Two types of autonomic nerves are present in the liver: sympathetic and parasympathetic. The sympathetic nerve is believed to be a postganglionic nerve originating from the celiac ganglion (CG) [17]. Although studies have shown that parasympathetic stimulation significantly attenuates HIRI by inhibiting hepatic inflammation, oxidative stress, and apoptosis [18], the effect of the sympathetic nerves on HIRI remains unclear and warrants further investigation.

Clock genes play a vital role in the generation of 24 h rhythms [19,20]. Growing evidence suggests that Baml1 (brain and muscle arnt-like) acts as a core clock gene that not only directly regulates circadian rhythmicity, but can also influence the expression of other clock genes [21]. Bmal1, also called Arntl, is a transcription factor with a bHLH/PAS domain that plays a core role in the transcription/translation feedback loop (TTFL) of the biological clock. Bmal1 forms a heterodimer with the protein product encoded by the clock gene and activates the transcription of genes with E-box elements, such as Per 1, Per 2, Per 3, Cry 1, and Cry 2. The PER and CRY proteins then inhibit CLOCK/BMAL1 heterodimer activity, forming a negative feedback loop [22,23].

Although studies have indicated the role of CG in modulating liver physiology through the sympathetic nervous system, the underlying mechanism remains unclear [16]. Yu et al., have shown that Bmal1 knockdown in the left stellate nerve suppresses neural activity and improves myocardial ischemia [24]. However, whether Bmal1 knockdown alleviates HIRI is still unclear. This study aims to investigate the expression of Bmal1 in the CG and its regulatory role and mechanism in HIRI.

## 2. Methods

### 2.1. Animal Experiments

Twelve adult male beagles (body weight: 10 ± 2 kg; age: 6 ± 1 months) provided by the Laboratory Animal Center of the Wuhan University School of Medicine were randomly assigned to two groups: the Bmal1 knockdown group (KO-Bmal1, N = 6) and the control group (control, N = 6). This study was conducted following the recommendations of the National Institutes of Health Guidelines for the Care and Use of Laboratory Animals. The Animal Experiment Ethics Committee of Wuhan University approved the protocol. All dogs were anesthetized using sodium pentobarbital (30 mg/kg) and ventilated with room air using a positive pressure respirator (MAO01746, Harvard Apparatus, Holliston, Massachusetts). An additional maintenance dose of 60 mg/h of sodium pentobarbital was administered during the surgery. Normal saline (100 mL/h) was injected to prevent spontaneous dehydration. Systemic arterial pressure was monitored through left femoral artery cannulation, and surface electrocardiograms (ECGs) were recorded using a computer-based laboratory system (Lead 7000, Jinjiang, China). A heating pad maintained the core body temperature at 36.5 ± 0.5 °C.

### 2.2. Bmal1 Knockdown and HIRI Modeling

In the KO-Bmal1 group, adeno-associated virus (AAV) (AAV2/PHPS-hSyn-mCherry-5′miR-30a-shRNA (Bmal1)-3′miR-30a-WPREs) was used as a genome editing tool, providing short hairpin RNAs (shRNAs) against Bmal1 into the CG. In contrast, the control group received AAV (AAV2/PHPS-mCherry-hSyn-5′miR-30a-shRNA (scramble)-3′miR-30a-WPREs). After four weeks, the dogs were anesthetized with pentobarbital (60 mg/kg; Sigma) and underwent open surgery along the lower edge of the rib cage. After extracting a small sample of liver tissue and serum, the portal vein and hepatic artery were clamped using vascular clips to interrupt the blood supply to the left or central lobe of the liver. After 1 h of ischemia, the vascular clamp was released, and the animals were euthanized after 6 h to collect liver tissue, other organ tissue, and serum samples for further analysis (Figure 1A,B).

### 2.3. Hematoxylin and Eosin (HE) Staining and Suzuki Scores

Liver tissues were fixed in 4% paraformaldehyde, embedded in paraffin, serially sectioned with a thickness of 4 μm, baked at 65 °C for 2 h, dewaxed, and hydrated in xylene. Sections were incubated in hematoxylin solution for 5 min and stained with eosin solution for 3 min. Subsequently, the sections were dehydrated, sealed, and viewed under a light microscope. Suzuki scores were used to evaluate liver tissue damage [25].

### 2.4. Immunohistochemical Staining and Immunofluorescence Staining

Liver sections were deparaffinized and rehydrated. Antigen retrieval was performed using an EDTA antigen retrieval buffer (pH 8.0) (Servicebio, Wuhan, China). A solution of 3% hydrogen peroxide and 3% bovine albumin (Servicebio, China) was used for blocking, and 0.2% Triton X-100 (Solarbio, China) was used for permeabilization. Sections were incubated with primary antibodies overnight at 4 °C (anti-Bax, Abways Technology, China; anti-TH, Proteintect, China; anti-cleaved caspase-3, Abways Technology, Shanghai, China; anti-TGF-β1, Abways Technology, China; anti-a-SMA, Boster Biological Technology co.ltd, China), followed by secondary antibodies at 37 °C for 1 h. The staining results were visualized using 3,5-diaminobenzidine (Servicebio, China).

As for immunofluorescence staining, an EDTA antigen retrieval buffer (pH 9.0) (Servicebio, China) was used for antigen retrieval, and 10% donkey serum was used for blocking. Samples were incubated overnight at 4 °C with the primary antibodies (anti-TH, Proteintect, USA; anti-c-fos, Abcam, Waltham, MA, USA; anti-NGF, Abcam, USA). Secondary antibody incubation was performed for 1 h at 37 °C, and nuclei were labelled with DAPI (Abcam, USA). The images were taken with a light microscope (Olympus Corporation).

### 2.5. RNA Isolation and Quantitative PCR

Total RNA was extracted from the livers of dogs using the Trizol reagent (Servicebio, China). Reverse transcription of total RNA was performed using a reverse transcription kit (ThermoFisher Scientific, Waltham, MA, USA). The product was used for subsequent quantitative real-time polymerase chain reaction (qRT-PCR) detection of Bmal1, TNF-α, and IL-1β (Servicebio, China). GAPDH was used as an internal reference for normalization. The primer sequences used for Bmal1 were as follows: (forward) 5′-AAGGGAAGCTCACAGTCAGAT and (reverse) 5′-GGACATTGCGTTGCATGTTGG. The primers used for TNF-α were as follows: (forward) 5′-CCTGGAGCCAAATTCGAGTG and (reverse) 5′-CGTGTGGGTTCTGTCTCGTG. The primers used for IL-1β were as follows: (forward) 5′-CCAGCTCAGAGGCTCTTGTG and (reverse) 5′-AGGGCCAGGTCCTGAGAAAT.

### 2.6. Western Blotting

The liver tissue was lysed in a RIPA buffer (Servicebio, China) that was supplemented with proteinase (Servicebio, China), phosphatase inhibitors (Servicebio, China), and PMSF (Servicebio, China). The lysates were then centrifuged at 12,000 rpm at 4 °C for minutes to remove debris, and the resulting supernatants were collected for electrophoresis. Western blotting was performed according to the protocol described in our previous study [26]. The primary antibodies used for the western blotting were TNF-α (Servicebio, China), IL-1β (Servicebio, China), and Tubulin-β (Abways Technology, Shanghai, China). Quantitative analysis of the western blotting was performed using ImageJ software.

### 2.7. MDA Detection

An MDA detection kit (Solarbio, Beijing, China) was used to detect MDA levels in liver tissues. For homogenization (60 Hz, 2 min) in an ice bath, 100 μL of the lysis buffer was added from the MDA kit per 10 mg of tissue. The corresponding reagents were added according to the manufacturer’s instructions, and the reaction was heated in a 100 °C water bath for 60 min. The reaction was then cooled on ice and centrifuged at 10,000× *g* for 10 min. The supernatant was collected and added to a 96-well plate, and the absorbance values were detected at 450 nm, 532 nm, and 600 nm on a microplate reader.

### 2.8. GSH Detection

The detection of GSH in the liver tissue was performed using a GSH detection kit (Jiancheng, Nanjing, China). The tissue was carefully and accurately weighed according to the ratio of weight (g): volume (mL) = 1:9, and normal saline was added to make the tissue homogenate. The sample was centrifuged at 2500 rpm for 10 min; next, the supernatant was collected, mixed with the reagent in a ratio of 1:1, and centrifuged at 3500 rpm for 10 min. Finally, the supernatant was collected, and the corresponding reagents were added according to the manufacturer’s instructions, mixed well, and left to stand for 5 min. Each well’s absorbance was measured at 405 nm by the microplate reader.

### 2.9. Periodic Acid Schiff (PAS) Staining, Oil Red O Staining, and Masson Trichrome Staining

Sections of paraffin-embedded liver tissue were stained with 0.5% periodic acid for 10 min, chevron dye for 15 min in the dark, and hematoxylin for 2 min using the PAS staining kit (Servicebio, China). Frozen sections were stained using the Oil Red O staining kit (Pinuofei, Wuhan, China) and counterstained with hematoxylin according to the manufacturer’s instructions. The ready-to-use kit (Pinuofei, China) was used to perform the Masson’s trichrome staining. Briefly, the sections were sequentially immersed in Weigert’s iron hematoxylin solution for 5 min, stained with Ponceau S for 10 min, treated with a phosphomolybdic acid solution for 5 min, and stained with an aniline blue solution for 5 min. Finally, the sections were separated using 1% glacial acetic acid for 1 min. Images were taken using a light microscope (Olympus Corporation).

### 2.10. Enzyme-Linked Immunosorbent Assay (ELISA)

Norepinephrine (NE) and neuropeptide Y (NPY) in the supernatant were measured using the enzyme-linked immunosorbent assay kit (Jianglaibio, Shanghai, China) according to the manufacturer’s instructions.

### 2.11. Serum Biochemistry Analysis

Blood samples were collected from the upper limbs of the Beagle dogs using a vacuum serum separator tube and left to clot naturally at room temperature for 30 min, followed by centrifugation at 1500× *g* for 10 min and collection of the supernatant serum. Serum alanine transaminase (ALT) and glutathione transaminase (AST) were measured using a fully automated biochemical instrument (Renmin Hospital of Wuhan University).

### 2.12. Statistical Analysis

All data are presented as mean ± SD. Student’s *t*-tests were used to analyze statistical differences. *p* < 0.05 was considered statistically significant.

## 3. Results

### 3.1. Evaluation of Bmal1 Knockdown Efficiency in CG

We first examined the anatomical location of the CG in the dogs (Figure 1C) and observed that the CG was enriched with ganglion cells and surrounded by satellite cells (Figure 1D,E). We injected CG with empty and suppressed Bmal1 AAV viruses in the two groups, respectively. To assess the efficiency of AAV transfection, we performed double immunofluorescence staining for tyrosine hydroxylase (TH) and mCherry (Figure 1F). TH was stained to label neurons, and mCherry was used as a marker protein for adeno-associated viruses. Quantitative analysis revealed high and identical efficiencies of viral transfection in CG in the Bmal1 knockout (KO-Bmal1) and control groups (Figure 1G). Additionally, the qRT-PCR results established that the expression of Bmal1 was significantly reduced in the KO-Bmal1 group (Figure 1H).

### 3.2. Knockdown of Bmal1 Decreases CG Neural Activity

c-fos and nerve growth factor (NGF) are classical markers of neural activity in the peripheral nervous system [27]. We examined the effect of Bmal1 knockdown on c-fos and NGF expression in the CG. Immunofluorescence staining of the CG demonstrated a significant reduction in the proportion of c-fos^+^ and NGF^+^ neurons in TH^+^ cells in the KO-Bmal1 group compared with the control group (Figure 2A,D), confirming that Bmal1 knockdown significantly decreased the neural activity in the CG.

### 3.3. Knockdown of Bmal1 Alleviates Liver Injury Caused by HIRI

The Suzuki scores were significantly higher in the HIRI beagles than in the non-HIRI beagles (Figure 3A,B). Bmal1 knockdown significantly reduced the Suzuki scores compared with the empty virus injection group (Figure 3A,B). Additionally, we observed a significant increase in serum ALT and AST in the HIRI beagles, whereas Bmal1 downregulation alleviated this increase (Figure 3C,D).

### 3.4. Downregulation of Bmal1 Increases Glycogen and Reduces Fat Accumulation

We performed PAS and Oil Red O staining to assess the effect of Bmal1 knockdown on hepatic glycogen and fat reserves. We found that HIRI resulted in a significant decrease in glycogen reserves and an increase in fat accumulation compared with the non-HIRI group (Figure 3E). In contrast, Baml1 knockdown significantly increased glycogen and decreased fat accumulation (Figure 3F).

### 3.5. Downregulation of Bmal1 Mitigates Hepatocyte Apoptosis Levels

Bax and cleaved caspase-3 are commonly used apoptosis marker proteins [28]. We estimated the expression of Bax and cleaved caspase-3 proteins in liver tissues to assess the effect of Bmal1 knockdown on hepatocyte apoptosis. We found that HIRI induced a significant upregulation in the expression of Bax and cleaved caspase-3 compared with the non-HIRI group (Figure 3G,H). In contrast, Baml1 knockdown significantly decreased the expression of Bax and cleaved caspase-3 in HIRI the beagle dogs (Figure 3G,H).

### 3.6. Knockdown of Bmal1 Attenuates HIRI-Induced Liver Fibrosis

The Masson Trichrome staining was used to examine the effect of Bmal1 knockdown on liver fibrosis. The results demonstrated that HIRI caused a significant proliferation of hepatic collagen fibers compared with the non-HIRI group (Figure 4A,B). Moreover, Baml1 knockdown significantly alleviated HIRI-induced liver fibrosis (Figure 4A,B).

We then assessed the expression of TGF-β1 and α-SMA proteins in the liver using immunohistochemical staining. We observed that HIRI significantly increased the expression of TGF-β1 and α-SMA compared with the non-HIRI group (Figure 4C,D). However, Baml1 knockdown in the HIRI beagle dogs dramatically lowered the expression of TGF-β1 and α-SMA (Figure 4E,F).

### 3.7. Down-Regulation of Bmal1 Inhibits Hepatic Neurotransmitter Levels and Sympathetic Nerve Activity in HIRI

As neurotransmitters are chemicals that transmit information between neurons or between neurons and effectors, we explored the effect of Bmal1 knockdown on hepatic neurotransmitter NE and NPY levels in the liver after HIRI. We found that the knockdown of Bmal1 significantly lowered NE and NPY compared with the empty vector control group (Figure 5A,B). The liver is a visceral organ regulated by sympathetic neurons in the CG; therefore, we labelled TH^+^ sympathetic nerves by immunohistochemical staining. The results indicated that HIRI caused sympathetic activation in the KO-Bmal1 and control groups, while Baml1 knockdown significantly reduced sympathetic activation (Figure 5C,D).

### 3.8. Decreased Bmal1 Expression in CG Alleviates Oxidative Stress and Inflammation Caused by HIRI

We evaluated TNF-α and IL-1β levels in the liver using qRT-PCR. We observed a significant reduction in the expression of TNF-α and IL-1β in the KO-Bmal1 group after HIRI compared with the control group (Figure 6A,B). Western blotting results further validated that Bmal1 knockdown significantly reduced the expression of TNF-α and IL-1β after HIRI (Figure 6C,D). These results indicated that the knockdown of Bmal1 significantly reduced the inflammatory response of the liver caused by HIRI.

Moreover, we detected the levels of GSH and MDA after HIRI to assess oxidative stress. We observed that the GSH level was significantly increased and the MDA level was significantly decreased in the KO-Bmal1 group compared with the control group (Figure 6E,F).

## 4. Discussion

This study demonstrated that the inhibition of Bmal1 in the CG can effectively protect dogs against HIRI by reducing hepatic sympathetic nervous activity. Firstly, we successfully constructed an animal model of Bmal1 knockdown by injecting an AAV into the CG. Secondly, we found that Bmal1 downregulation in CG reduced liver injury by inhibiting neural activity in CG which led to decreased hepatic neurotransmitter levels, hepatic neural activity, hepatic inflammatory response, and oxidative stress levels. The choice of beagles as the animal model was based on their close biological rhythm to humans compared with rodents. This study highlights the endogenous circadian clock system of the autonomic nervous system and suggests that Bmal1, a key regulator of circadian rhythms, may have a potential role in the prevention and treatment of HIRI.

The liver is primarily innervated by sympathetic nerves [29]. Several studies have confirmed that neurogenic signals affect the IRI process, such as activation of the vagus nerve to inhibit HIRI; however, the role of sympathetic nerves in HIRI remains unclear [18,30]. Friman et al. [31] used guanethidine to inhibit sympathetic nerves in rats and observed no significant differences in ALT and mean arterial pressure between liver IRI and control rats. However, the experimental design was simple with a small data set, thus requiring further verification. Furthermore, the CG is an important ganglion for the sympathetic regulation of liver function in humans and mice [32,33], although there are currently no reports regarding dogs. Thus, we first identified the anatomical location of CG and the expression of Bmal1 in CG in dogs. Bmal1 downregulation in CG using AAV significantly reduced neurotransmitter levels and sympathetic neural activity of the liver, suggesting that CG may regulate liver circadian rhythms.

In this study, the inhibition of sympathetic nerve activity improved inflammation after HIRI. Our results reveal that Bmal1 inhibition reduces sympathetic nerve activity in the CG and the liver. We established HIRI models by clipping and releasing the portal vein and hepatic artery in dogs in both groups. HE staining and Suzuki scores demonstrated that the HIRI was less severe in the KO-Bmal1 group than in the control group, which confirmed that the inhibition of sympathetic nerve activity has a protective effect on HIRI. The protein expression of p-Akt and Bmal1 was significantly increased in the hearts of myocardial ischemic mice. A BDNF mimetic, 7,8-dihydroxyflavone (7,8-DHF), alleviates cardiac fibrosis by restoring circadian signals via downregulating the Bmal1/Akt pathway [34]. Sun et al., found that the expression levels of clock and Bmal1 proteins were significantly higher after 48 h of cerebral hypoxia and ischemia [35]. Lembach et al., demonstrated that the area of the brain infarct core in Bmal1 knockout female mice was significantly smaller than that in wild-type females at 14 days after photothrombosis, confirming that Bmal1 knockout plays an active role in brain infarction [36]. Interestingly, Bmal1 knockdown in the left stellate ganglion leads to changes in the transcriptional levels of genes associated with neural activity, thereby preventing ventricular arrhythmias after myocardial ischemia [24]. These studies, therefore, reveal that ischemia leads to Bmal1 upregulation, while Bmal1 downregulation alleviates ischemia-induced organ damage, which is consistent with our findings.

HIRI involves a variety of inflammatory cytokine-related molecular mechanisms [37]. We established that suppressed hepatic sympathetic nerve activity in HIRI caused by Bmal1 downregulation in the CG resulted in decreased liver fibrosis and inflammatory cytokine levels, including IL-1β and TNF-α. Recent studies have revealed the crucial role of the circadian rhythm in the production and release of collagen in fibroblasts [38]. Furthermore, a disrupted circadian rhythm has been linked to the development of various fibrotic disorders. Zhang et al., demonstrated that Bmal1 levels increased in cultured murine proximal tubular cells following TGF-β treatment and that Bmal1 deficiency protected mice against obstructive renal fibrosis [39]. Similarly, Hang et al., also found that the downregulation of Bmal1 attenuated cardiac fibrosis by inhibiting the Akt pathway [34]. Consistent with these findings, our study also demonstrated that Bmal1 downregulation significantly inhibited liver fibrosis caused by HIRI. IL-1β, a cytokine that promotes HIRI, is essential for initiating inflammatory responses [40]. Kamo et al. [41] demonstrated that the neutralization of IL-1β using a monoclonal antibody attenuated HIRI in mice. Additionally, we investigated the effect of oxidative stress on HIRI. While oxidative stress plays a crucial role in the early stages of hepatic ischemia-reperfusion, the activation of multiple inflammatory pathways ultimately leads to the accumulation of neutrophils in the liver [42]. Neutrophils attack hepatocytes by releasing oxidants which are responsible for the later stages of IRI-induced liver injury [43]. The significant decrease in serum ROS in the KO-Bmal1 group reflects their lower levels of oxidative stress. Furthermore, GSH has a protective effect against inflammation-induced oxidative stress injury, which also explains the increase in GSH in the KO-Bmal1 group [44].

This study has certain limitations. As a clock gene, Bmal1 is closely related to biological rhythms. All our experiments started at 8:00 a.m.; however, we did not set up a nighttime experiment as a control. Therefore, this experiment may explain the role of Bmal1 in reducing sympathetic activity in HIRI injury but not the role of biological rhythms in HIRI. It would be interesting to assess the long-term effects of Bmal1 knockdown on longer reperfusion times. Further studies are needed to determine the underlying mechanisms of action of Bmal1 in the CG and HIRI.

## 5. Conclusions

In conclusion, our findings suggest that Bmal1 knockdown in the CG can ameliorate hepatocyte injury induced by HIRI, likely via the inhibition of hepatic sympathetic nervous activity. These results suggest that targeting Bmal1 in the CG may have therapeutic potential for the treatment of HIRI. Further studies are needed to explore the underlying mechanisms and to develop drugs that target Bmal1 in CG for clinical use.

## Figures and Tables

**Figure 1 biomolecules-13-00713-f001:**
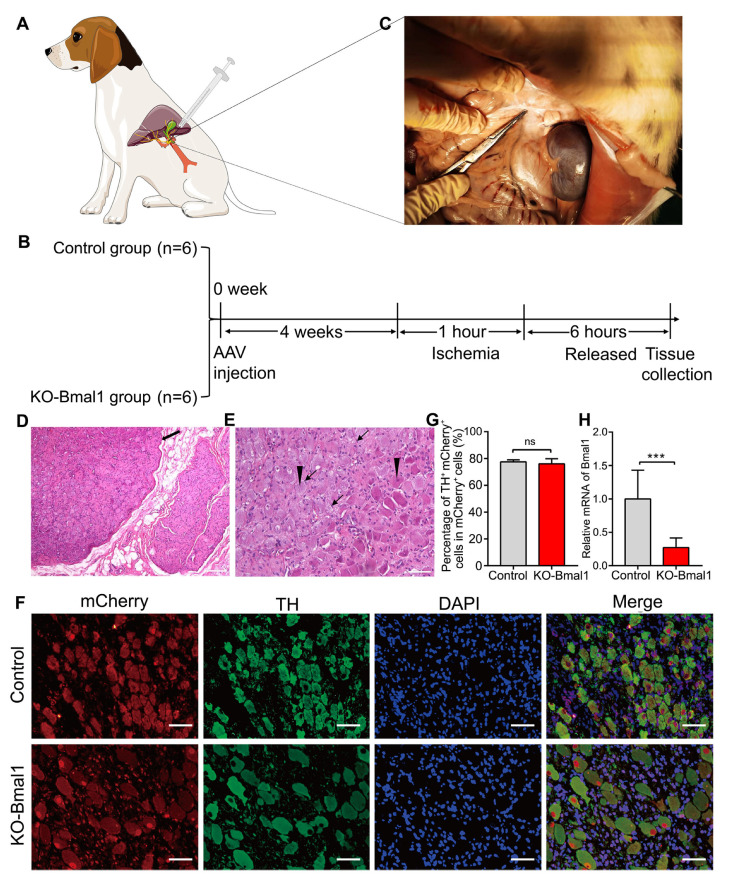
Evaluation of Bmal1 knockdown in the celiac ganglia. (**A**) Schematic diagram of virus microinjection in the celiac ganglia. (**B**) Flowchart of the experiment. (**C**) The anatomical location of the celiac ganglia. (**D**,**E**) HE staining of the celiac ganglia (**D**, 40×; **E**, 200×). The thick arrows represent ganglia, the thin arrows represent satellite cells, and the triangles represent nodal cells. (**F**,**G**) The percentage of TH^+^ mCherry^+^ neurons in mCherry^+^ cells (400×). (**H**) The expression of Bmal1 was estimated by quantitative RT-PCR. TH, tyrosine hydroxy; AAV, adeno-associated virus. Student’s *t*-tests were used to analyze statistical differences, and each group had six beagles. *** *p* < 0.001. ^ns^
*p* > 0.05.

**Figure 2 biomolecules-13-00713-f002:**
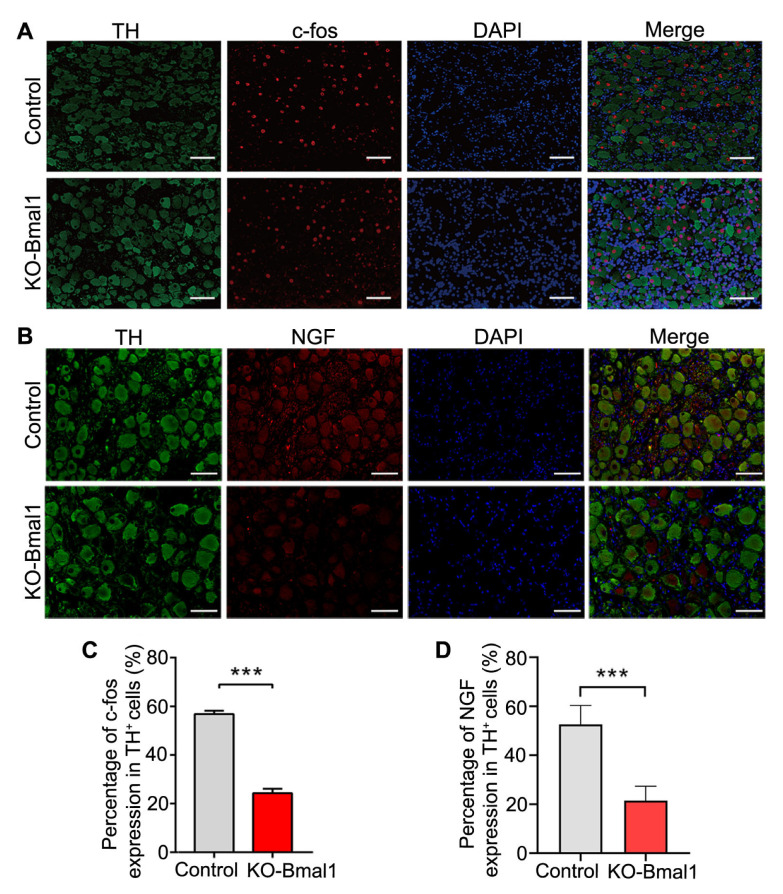
Impact of c-fos and NGF expression in the celiac ganglia after Bmal1 knockdown. (**A**,**B**) Representative immunofluorescent co-staining images for c-fos and NGF with TH in the celiac ganglia (400×). (**C**,**D**) Quantitative analysis of c-fos and NGF expressed as the percentage of TH^+^ cells. NGF, nerve growth factor; TH, tyrosine hydroxy. Student’s *t*-tests were used to analyze statistical differences, and each group had six beagles. *** *p* < 0.001.

**Figure 3 biomolecules-13-00713-f003:**
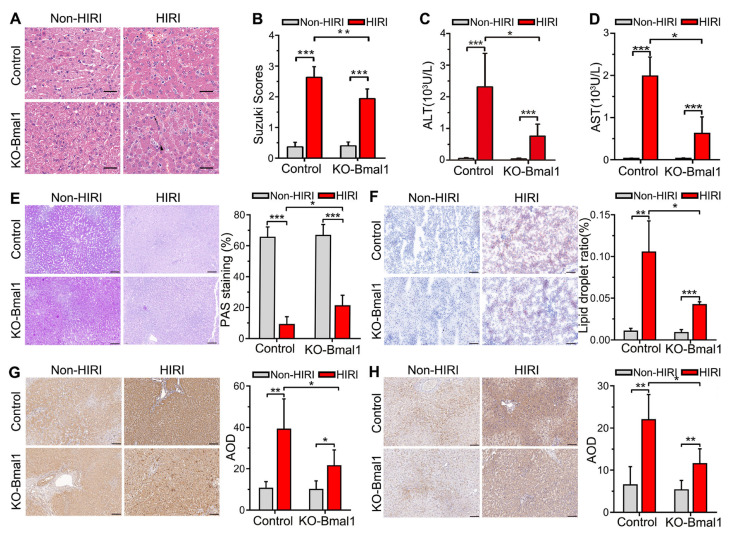
Decreased expression of Bmal1 in the celiac ganglia alleviates hepatic ischemia-reperfusion injury. (**A**,**B**) HE staining and Suzuki scores in the liver. The arrows represent inflammatory cell infiltration, and the triangle represents liver cell necrosis. (**C**,**D**) Analysis of serum ALT and AST levels. (**E**) Representative image of PAS staining of liver sections (200×) and their quantitative analysis results. (**F**) Representative image of Oil Red O staining of liver sections (200×) and their quantitative analysis results. (**G**) Representative image of immunohistochemical staining of cleaved caspase 3 in liver sections (200×) and their quantitative analysis results. (**H**) Representative image of immunohistochemical staining of BAX in liver sections (200×) and their quantitative analysis results. HIRI, hepatic ischemia-reperfusion injury; AST, aspartate aminotransferase; AST, alanine aminotransferase; AOD, average optical density. Student’s *t*-tests were used to analyze statistical differences, and each group had six beagles. * *p* < 0.05, ** *p* < 0.01, and *** *p* < 0.001.

**Figure 4 biomolecules-13-00713-f004:**
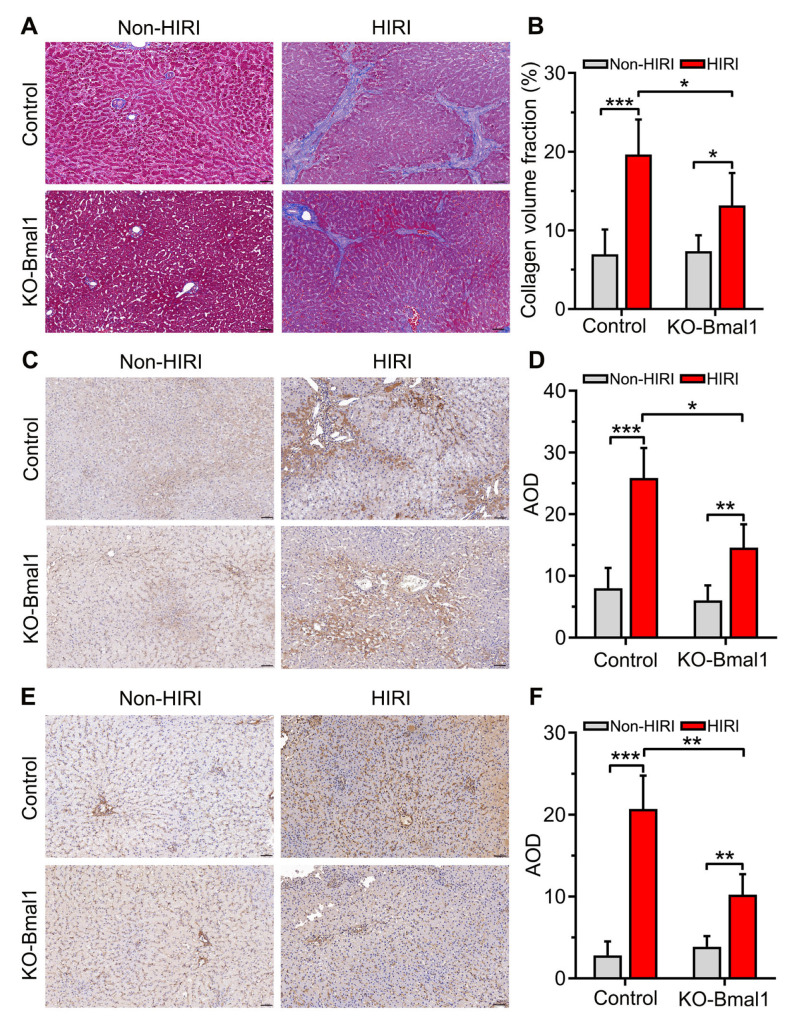
Reduced expression of Bmal1 in the celiac ganglia attenuates liver fibrosis in hepatic ischemia-reperfusion injury. (**A**) Representative image of Masson Trichrome staining of liver sections (200×). (**B**) The quantitative analysis of Masson Trichrome staining results. (**C**) Representative image of immunohistochemical staining of TGF-β1 in liver sections (200×). (**D**) The quantitative analysis of TGF-β1 expression. (**E**) Representative image of immunohistochemical staining of α-SMA in liver sections (200×). (**F**) The quantitative analysis of α-SMA expression. HIRI, hepatic ischemia-reperfusion injury; AOD, average optical density. Student’s *t*-tests were used to analyze statistical differences, and each group had six beagles. * *p* < 0.05, ** *p* < 0.01, and *** *p* < 0.001.

**Figure 5 biomolecules-13-00713-f005:**
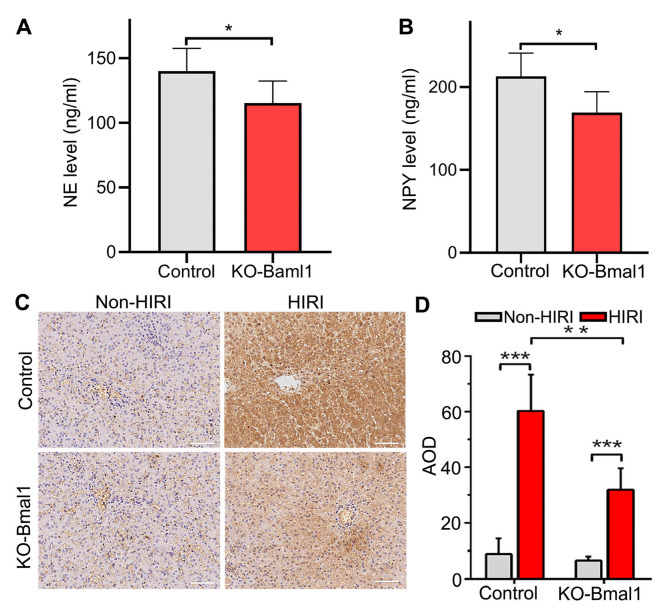
Down-regulation of Bmal1 expression in the celiac ganglia inhibits hepatic neurotransmitter levels and sympathetic nerve activity in hepatic ischemia-reperfusion injury. (**A**,**B**) Quantification of hepatic NE (**A**) and NPY (**B**) levels detected by ELISA. (**C**) Representative image of immunohistochemical staining showing TH in liver sections (200×). (**D**) The quantitative analysis of TH expression. NE, norepinephrine; NPY, neuropeptide Y; TH, tyrosine hydroxy; AOD, average optical density; HIRI, hepatic ischemia-reperfusion injury. Student’s *t*-tests were used to analyze statistical differences, and each group had six beagles. * *p* < 0.05, ** *p* < 0.01, and *** *p* < 0.001.

**Figure 6 biomolecules-13-00713-f006:**
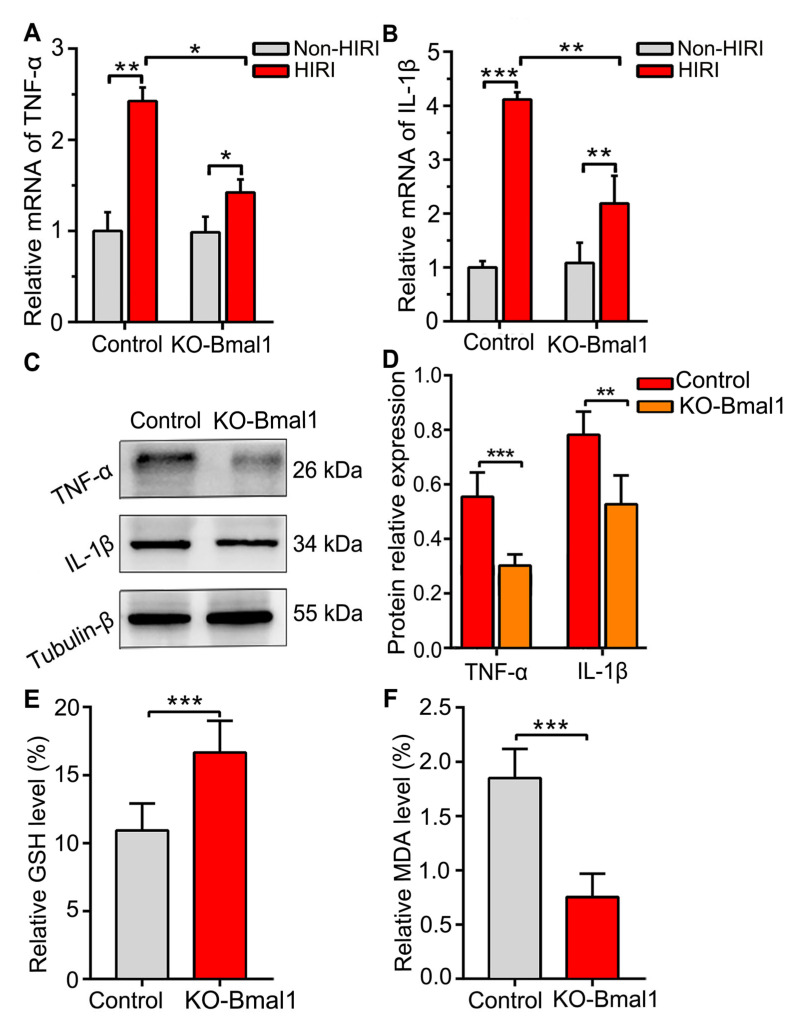
Decreased Bmal1 expression in the celiac ganglia inhibits oxidative stress and inflammation in hepatic ischemia-reperfusion injury. (**A**–**D**) Analysis of the expression levels of TNF-α and IL-1β in the liver by qRT-PCR (**A**,**B**) and western blotting (**C**,**D**). (**E**,**F**) Analysis of GSH and MDA levels in the liver. Student’s *t*-tests were used to analyze statistical differences, and each group had six beagles. HIRI, hepatic ischemia-reperfusion injury; GSH, glutathione; MDA, malondialdehyde. * *p* < 0.05, ** *p* < 0.01, and *** *p* < 0.001.

## Data Availability

The datasets used and analyzed during the present study are available from the corresponding author on reasonable request.

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
