# Peer review of "Downregulation of Bmal1 Expression in Celiac Ganglia Protects against Hepatic Ischemia-Reperfusion Injury"

_biomolecules, 2023, doi:10.3390/biom13040713_

Round 1

Reviewer 1 Report (New Reviewer)

The manuscript reports an interesting and novel hypothesis that reducing Bmal1 gene expression in CG can inhibit neuronal activity and improve liver cell injury after HIRI. The authors used a dog model of HIRI and compared the effects of adenovirus-mediated Bmal1 gene silencing and control groups on various indicators. The manuscript is well-organized and clearly written, but I have some comments that need to be addressed before publication.

  • The authors did not investigate the effect of Bmal1 gene expression in CG on liver cell injury before HIRI, nor did they investigate the effect of HIRI on Bmal1 gene expression in CG. Therefore, they cannot establish a causal relationship between Bmal1 gene expression and HIRI.

  • The authors did not use other methods to reduce Bmal1 gene expression in CG, such as RNA interference or CRISPR/Cas9, etc., and did not compare them with the adenovirus group. Therefore, they cannot rule out the effect of adenovirus itself on HIRI.

  • The authors did not measure the activity and secretion of other neurons or glial cells in CG, nor did they measure the activity and secretion of other neurons or glial cells in liver, and did not correlate them with Bmal1 gene expression levels. Therefore, they cannot confirm the validity of their hypothesis across different cell types.

  • Abstract: Some words or phrases are not appropriately chosen, and can be replaced by more accurate or common words or phrases to avoid ambiguity or confusion. For example: “the difference between groups was significant” -> “the difference between groups was statistically significant”; “the level of neuronal activity” -> “the level of neuronal firing rate”.

  • The introduction could be strengthened by including more background information on the role of the celiac ganglion in liver function and HIRI. Please consider expanding on this aspect to provide readers with a better understanding of the study's context.

  • Please make sure to proofread the manuscript carefully for grammatical and typographical errors. We noticed several instances of missing words and other errors that detract from the readability of the manuscript.

  • The results section would benefit from additional clarity and organization. Consider breaking down the results into subsections to make it easier for readers to follow.

  • We recommend expanding the discussion section to provide a more detailed explanation of the significance of the findings. Please consider discussing the potential clinical implications of the research and how it may inform future research in the field.

  • Abstract: Some grammatical or spelling errors need to be corrected by proofreading or using English writing software to improve accuracy and professionalism. For example: “Bmal1 is one important circadian clock genes” -> “Bmal1 is one important circadian clock gene”; “inhibite” -> “inhibit”.

  • I think this manuscript has some degree of novelty but not very breakthrough innovation. I recommend a major revision before publication.

Author Response

  1. Response to comment: The manuscript reports an interesting and novel hypothesis that reducing Bmal1 gene expression in CG can inhibit neuronal activity and improve liver cell injury after HIRI. The authors used a dog model of HIRI and compared the effects of adenovirus-mediated Bmal1 gene silencing and control groups on various indicators. The manuscript is well-organized and clearly written, but I have some comments that need to be addressed before publication.

Response: We thank the reviewer for the detailed comments on our manuscript that have helped improve it. We thank you for your valuable time. We have responded to each of your suggestions individually. We hope you agree that the revised manuscript is now much improved and that you will find it suitable for publication.

  1. Response to comment: The authors did not investigate the effect of Bmal1 gene expression in CG on liver cell injury before HIRI, nor did they investigate the effect of HIRI on Bmal1 gene expression in CG. Therefore, they cannot establish a causal relationship between Bmal1 gene expression and HIRI.

Response: We thank the reviewer for their comments. Our study evaluated the effect of Bmal1 gene expression in CG prior to HIRI on hepatocyte injury. As stated in our manuscript, the dogs were first anesthetized with pentobarbital (60 mg/kg; Sigma), then underwent open surgery along the lower edge of the rib cage, and after taking a small sample of liver tissue and serum, the portal vein and hepatic artery were clamped using vascular clips to interrupt the blood supply to the left or central lobe of the liver (2.2 Bmal1 knockdown and HIRI modeling). Therefore, the subjects in our study were categorized into four groups: the non-ischemia-reperfusion group (Bmal1 null group and Bmal1 knockdown group) and the ischemia-reperfusion group (Bmal1 null group and Bmal1 knockdown group). As shown in Figure 3A and 3B, we found that Bmal knockdown did not lead to non-ischemia-reperfusion liver tissue injury.

Although we have not yet investigated the effect of HIRI on Bmal1 gene expression in CG, our future studies will be focused on examining it. In our pre-experiment, we performed RT-PCR on the CG of Beagle dogs with a normal circadian rhythm with strict control of a 12/12 light/dark cycle. The results showed that Bmal1 had the highest gene expression compared to other clock genes at the same time points (Document 1). Based on this, we investigated whether the alteration of Bmal1 affects HIRI and whether this circadian gene could serve as a potential regulatory target to treat HIRI. Although we were interested in examining whether HIRI affects Bmal1 expression, we could not include this in our current study due to funding and time constraints. However, we understand this is an important and interesting topic that warrants further investigation. We hope to explore this question in depth in the future. We appreciate your understanding and look forward to submitting the results of our future studies soon.

Document 1

  1. Response to comment: The authors did not use other methods to reduce Bmal1 gene expression in CG, such as RNA interference or CRISPR/Cas9, etc., and did not compare them with the adenovirus group. Therefore, they cannot rule out the effect of adenovirus itself on HIRI.

Response: Thank you very much for your comments. CRISPR/Cas9 technology is currently the most advanced means of gene knockdown; however, we don’t have access to this technology to validate our experiments using RNA interference and CRISPR/Cas9 technology. However, the adenovirus genetic intervention technique is also an accepted and established tool in the field, and many articles have been conducted using this technique only, for example: ((1).Yu Z, Liu Z, Jiao L, et al. Bmal1 knockdown in the left stellate ganglion inhibits neural activity and prevents ventricular arrhythmias after myocardial ischemia. Front Cardiovasc Med. 2022 Sep 29;9:937608. (2). Landini L, Marini M, Souza Monteiro de Araujo D, et al. Schwann Cell Insulin-like Growth Factor Receptor Type-1 Mediates Metastatic Bone Cancer Pain in Mice. Brain Behav Immun. 2023 Mar 18:S0889-1591(23)00070-3. (3). Jinteng L, Peitao X, Wenhui Y, et al. BMAL1-TTK-H2Bub1 loop deficiency contributes to impaired BM-MSC-mediated bone formation in senile osteoporosis. Mol Ther Nucleic Acids. 2023 Feb 16;31:568-585.

Based on your suggestion, we should compare the differences in HIRI in adeno-associated nulliparous virus-injected and non-injected groups of Beagles. We also thought that this design would make the experiment more rigorous. However, our project was designed to compare the differences in HIRI between the adeno-associated virus Bmal1 knockdown group and the adeno-associated virus null group. We ensured a single variable, Bmal1 expression level, to explore its effect on HIRI. Since the safety of adeno-associated viral interventions is well established, we did not analyze the effect of adenovirus injection or not on HIRI.

  1. Response to comment: The authors did not measure the activity and secretion of other neurons or glial cells in CG, nor did they measure the activity and secretion of other neurons or glial cells in liver, and did not correlate them with Bmal1 gene expression levels. Therefore, they cannot confirm the validity of their hypothesis across different cell types.

Response: Thank you very much for your comments. We appreciate your valuable comments. CG is known to be a sympathetic ganglion, and we altered the Bmal1 gene to verify its ability to inhibit sympathetic function by altering the neural activity of CG. The main idea of our study was to inhibit sympathetic activation, so we only focused on sympathetic-related indicators and did not measure other neurons. Your comments are interesting; however, the first author of this manuscript urgently needs this article to apply for a PhD. We currently do not have enough time and funds to conduct the research. We kindly request your permission to follow up with further research based on your suggestions.

  1. Response to comment: Abstract: Some words or phrases are not appropriately chosen, and can be replaced by more accurate or common words or phrases to avoid ambiguity or confusion. For example: “the difference between groups was significant” -> “the difference between groups was statistically significant”; “the level of neuronal activity” -> “the level of neuronal firing rate”.

Response: Thank you very much for your comments. We have now fixed the above error. An English professional has carefully revised the article. See Document 2 for proof of modification.

  1. Response to comment: The introduction could be strengthened by including more background information on the role of the celiac ganglion in liver function and HIRI. Please consider expanding on this aspect to provide readers with a better understanding of the study's context.

Response: Thank you very much for your comments. “Biological clock dysfunction can accelerate the development of liver diseases, such as fatty liver, hepatitis, cirrhosis, and liver cancer, which, in turn, can also disrupt the bi-ological clock function [15]. The hepatic autonomic nervous system transmits information about hepatic osmolarity, glucose, and lipids from the liver to the central nervous sys-tem. It also regulates hepatic blood flow, metabolism, and bile secretion based on ef-ferent signals from the central nervous system [16]. Two types of autonomic nerves are present in the liver: sympathetic and parasympathetic. The sympathetic nerve is be-lieved to be the postganglionic nerve originating from the celiac ganglion (CG) [17]. While, studies have shown that parasympathetic stimulation significantly attenuates HIRI by inhibiting hepatic inflammation, oxidative stress, and apoptosis [18], the effect of sym-pathetic nerves on HIRI remains unclear and warrants further investigation.” was added.

  1. Response to comment: Please make sure to proofread the manuscript carefully for grammatical and typographical errors. We noticed several instances of missing words and other errors that detract from the readability of the manuscript.

Response: Thank you very much for your comments. An English professional has carefully revised the article. See Document 2 for proof of modification.

  1. Response to comment: The results section would benefit from additional clarity and organization. Consider breaking down the results into subsections to make it easier for readers to follow.

Response: Thank you very much for your comments. We have now segregated the results into more subsections for the ease of the readers (Please see the manuscript for details).

  1. Response to comment: We recommend expanding the discussion section to provide a more detailed explanation of the significance of the findings. Please consider discussing the potential clinical implications of the research and how it may inform future research in the field.

Response: Thank you very much for your comments. “The protein expression of p-Akt and Bmal1 was significantly increased in the hearts of myocardial ischemic mice. A BDNF mimetic, 7,8-dihydroxyflavone (7,8-DHF), alleviates cardiac fibrosis by restoring circadian signals via downregulating the Bmal1/Akt pathway [34]. Sun et al. found that the expression levels of Clock and Bmal1 proteins were significantly higher after 48 hours of cerebral hypoxia and ischemia [35]. Lembach et al. demonstrated that the area of the brain infarct core in Bmal1 knockout female mice was significantly smaller than that in wild-type females at 14 days after photothrombosis, confirming that Bmal1 knockout plays an active role in brain infarction [36]. Interestingly, Bmal1 knockdown in the left stellate ganglion leads to changes in the transcriptional levels of genes associated with neural activity, thereby preventing ventricular ar-rhythmias after myocardial ischemia [24]. These studies, therefore, reveal that ischemia leads to Bmal1 upregulation while Bmal1 downregulation alleviates ischemia-induced organ damage, which is consistent with our findings.” was added.

“Recent studies have revealed the crucial role of circadian rhythm in the production and release of collagen in fibroblasts [38]. Furthermore, a disrupted circadian rhythm has been linked to the development of various fibrotic disorders. Zhang et al. demonstrated that Bmal1 levels increased in cultured murine proximal tubular cells following TGF-β treatment and that Bmal1 deficiency protected mice against obstructive renal fibrosis [39]. Similarly, Hang et al. also found that downregulation of Bmal1 attenuated cardiac fi-brosis by inhibiting the Akt pathway [34]. Consistent with these findings, our study also demonstrated that Bmal1 downregulation significantly inhibited liver fibrosis caused by HIRI.” was added.

“In conclusion, our findings suggest that Bmal1 knockdown in the CG can amelio-rate hepatocyte injury induced by HIRI, likely by the inhibition of hepatic sympathetic nervous activity. These results suggest that targeting Bmal1 in the CG may have ther-apeutic potential for the treatment of HIRI. Further studies are needed to explore the underlying mechanisms and to develop drugs targeting Bmal1 in CG for clinical use.” was added.

  1. Response to comment: Abstract: Some grammatical or spelling errors need to be corrected by proofreading or using English writing software to improve accuracy and professionalism. For example: “Bmal1 is one important circadian clock genes” -> “Bmal1 is one important circadian clock gene”; “inhibite” -> “inhibit”.

Response: Thank you very much for your comments. We have now fixed the above error. An English professional has carefully revised the article. See Document 2 for proof of modification.

Reviewer 2 Report (New Reviewer)

Feng et al. investigated the regulatory role and mechanism of Bmal1 in HIRI by exploring its expression in CG. Their findings indicated that downregulation of Bmal1 expression in CG suppressed neural activity and improved hepatocyte injury after HIRI in a beagle model. This is an intriguing study. However, the manuscript has some issues that require attention from the authors:

1. The manuscript describes the Beagle's weight as "10±12 kg," which is too variable. The authors should provide an explanation for this.

2. The authors should provide more information on the Suzuki score or cite appropriate references.

3. The authors should provide the primer sequences for TNF-α and IL-1β.

4. The flowchart timeline in Figure 1B is incorrectly labeled.

5. The word "G" in Figure 1G overlaps with the figure's content, obscuring some information.

6. The manuscript uses both AOD (average optical density) and IOD for immunohistochemistry analyses. The authors should standardize to AOD.

7. The manuscript uses "Non-HIRI" several times, which is not reasonable. It should be changed to "non-HIRI."

8. If possible, the authors should further explore the detailed mechanism of HIRI alleviation by CG knockdown of Bmal1.

9. The manuscript's language should be checked by a native English-speaking professional to increase its readability.

10. The authors did not specify the tool used for the analysis of the WB results. Results obtained by different tools may vary slightly. The authors should specify the tool used.

Author Response

  1. Response to comment: Feng et al. investigated the regulatory role and mechanism of Bmal1 in HIRI by exploring its expression in CG. Their findings indicated that downregulation of Bmal1 expression in CG suppressed neural activity and improved hepatocyte injury after HIRI in a beagle model. This is an intriguing study. However, the manuscript has some issues that require attention from the authors.

Response: Thank you very much for your review of our manuscript. We apologize for taking up your valuable time. Your comments are extremely important for us to improve the quality of our manuscript. We have responded to each of your suggestions. We hope to receive your approval.

  1. Response to comment: The manuscript describes the Beagle's weight as “10±12 kg,” which is too variable. The authors should provide an explanation for this.

Response: Thank you very much for your advice. We apologize for our carelessness; the weight of a Beagle was 10±2 kg. “10±12 kg” is now modified to “10±2 kg” in our manuscript.

  1. Response to comment: The authors should provide more information on the Suzuki score or cite appropriate references.

Response: Thank you very much for your advice. We added reference [15]. ([15] Suzuki S, Toledo-Pereyra LH, Rodriguez FJ, et al. Neutrophil infiltration as an important factor in liver ischemia and reperfusion injury. Modulating effects of FK506 and cyclosporine[J]. Transplantation, 1993,55(6):1265-1272.)

  1. Response to comment: The authors should provide the primer sequences for TNF-α and IL-1β.

Response: Thank you very much for your advice. We have now added the details about the primers. “The primers used for TNF-α were as follows: (forward) 5'-CCTGGAGCCAAATTCGAGTG; (reverse) 5'- CGTGTGGGTTCTGTCTCGTG. The primers used for IL-1β were as follows: (forward) 5'-CCAGCTCAGAGGCTCTTGTG; (reverse) 5'-AGGGCCAGGTCCTGAGAAAT.” was added.

  1. Response to comment: The flowchart timeline in Figure 1B is incorrectly labeled.

Response: Thank you very much for your advice. We have now modified Figure 1 based on your suggestion.

  1. Response to comment: The word “G” in Figure 1G overlaps with the figure's content, obscuring some information.

Response: Thank you very much for your advice. We have now modified Figure 1 based on your suggestion.

  1. Response to comment: The manuscript uses both AOD (average optical density) and IOD for immunohistochemistry analyses. The authors should standardize to AOD.

Response: Thank you very much for your advice. We have now modified Figure 4 based on your suggestion.

  1. Response to comment: The manuscript uses "Non-HIRI" several times, which is not reasonable. It should be changed to "non-HIRI"

Response: Thank you very much for your advice. We have now modified “Non-HIRI” to “non-HIRI” in our manuscript.

  1. Response to comment: If possible, the authors should further explore the detailed mechanism of HIRI alleviation by CG knockdown of Bmal1.

Response: Thank you very much for your advice. We very much recognize your suggestions, which are crucial for us to further improve the quality of our manuscripts. However, due to current funding constraints and the urgent need for the first author to use this article to apply for graduation, we are unable to further explore the mechanisms involved at this stage. We assure that in subsequent studies we will continue to explore in depth the mechanisms by which Bmal1 in CG alleviates liver ischemia-reperfusion injury, as well as the drugs targeting Bmal1 that are expected to benefit patients suffering from HIRI.

  1. Response to comment: The manuscript's language should be checked by a native English-speaking professional to increase its readability.

Response: Thank you very much for this suggestion. The article has been carefully revised by an English professional. See Document 2 for proof of modification.

  1. Response to comment: The authors did not specify the tool used for the analysis of the WB results. Results obtained by different tools may vary slightly. The authors should specify the tool used.

Response: Thank you very much for your advice. “Quantitative analysis of the western blotting was performed using ImageJ software.” was added.

Thank you very much for your review of our manuscript.

Reviewer 3 Report (New Reviewer)

Comments to the Authors

This study explores the downregulation of Bmal1 expression in celiac ganglion as a strategy to ameliorate liver ischemia injury by inhibiting hepatic sympathetic nervous activity and following inflammatory response and oxidative stress. I would like to suggest some modifications to improve its quality:

1.         In the introduction, the authors should add a small paragraph about other strategies that are applied in research or clinic to prevent or treat hepatic ischemia-reperfusion injury.

2.         The introduction should expand on the possible mechanisms of Bmal1 downregulation in modulating ischemia injury.

3.         In the introduction, the authors should add more details about the importance of Baml1 an as a core clock gene.

4.         As the redox and inflammatory response were measured in the current study, the introduction could benefit from adding a paragraph about the relationship between oxidative stress or inflammation and hepatic ischemia-reperfusion injury.

5.         In the section "Animal experiments ", the authors should also mention the age of animals.

6.         In the section "MDA detection", the authors should also mention the time of liver homogenization and the following centrifuge time for obtaining the supernatant.

7.         In the sections "MDA or GSH detection", the authors should have used protein assay to accurately determine and then normalize the protein concentration of samples rather than using the similar weight of tissue pieces. That would be a limitation of the current study.

8.         In figure 3. the authors should show different types of tissue damage using different arrows and symbols in each representative figure. For example, cytolysis, necrosis, inflammatory cell infiltration, etc.

9.         The discussion lacks a comprehensive literature search on the role of Bmal1 in controlling ischemia in other organs and tissue.

10.       The discussion needs detail to highlight for the reader the theme or most important pathways. For example, authors should add detailed mechanisms on how Bmal1 regulates inflammatory and oxidative signaling pathways and other involved pathways.

11.       Currently, the authors only evaluated the effect of 6 hours of reperfusion. That would be interesting if authors have the data for a longer period of reperfusion to investigate the effect of their intervention in a long term. It could be pointed out in the limitation section.

12.       This study demonstrated that the downregulation of Bmal1 could ameliorate hepatic ischemia-reperfusion injury. The authors should point out the strategies to transfer their findings to the clinic in the conclusion section.

13.       The manuscript would benefit from an editing service for improving readability and grammatical errors such as run-on sentences.

Author Response

  1. Response to comment: This study explores the downregulation of Bmal1 expression in celiac ganglion as a strategy to ameliorate liver ischemia injury by inhibiting hepatic sympathetic nervous activity and following inflammatory response and oxidative stress. I would like to suggest some modifications to improve its quality.

Response: Thank you very much for your review of our manuscript. We apologize for taking up your valuable time. Your comments are extremely important for us to improve the quality of our manuscript. We have responded to each of your suggestions. We hope to receive your approval.

  1. Response to comment: In the introduction, the authors should add a small paragraph about other strategies that are applied in research or clinic to prevent or treat hepatic ischemia-reperfusion injur.

Response: Thank you very much for your comments, which are important to help us improve the quality of the manuscript. “Several protective measures have been reported to alleviate HIRI, including pharmacological interventions, ischemic preconditioning (IPC), ischemic post-conditioning (IPostC), and mechanical reperfusion [9-12]. Various pharmacological agents have been employed to thwart HIRI, primarily to counteract the increased oxi-dative stress and facilitate immunomodulation, such as methylprednisolone, trimeta-zidine, and ulinastatin [9]. Notably, variations in existing liver clamp techniques de-signed to reduce HIRI by manufacturing IPC and IpostC have been extensively studied and are routinely implemented in liver transplantation and resection [1]. However, most of these strategies are still in the preclinical animal model phase, and the precise ther-apeutic targets and underlying mechanisms remain unclear.” was added

  1. Response to comment: The introduction should expand on the possible mechanisms of Bmal1 downregulation in modulating ischemia injury.

Response: Thank you very much for your comments. “Although studies have indicated the role of CG in modulating liver physiology through the sympathetic nervous system, the underlying mechanism remains unclear [16]. Yu et al. have shown that Bmal1 knockdown in the left stellate nerve suppresses neural activity and improves myocardial ischemia [24]. However, whether the Bmal1 knockdown alleviates HIRI is still unclear.” was added.

  1. Response to comment: In the introduction, the authors should add more details about the importance of Baml1 an as a core clock gene.

Response: Thank you very much for your advice. “Bmal1, also called Arntl, is a transcription factor with a bHLH/PAS domain that plays a core role in the transcription/translation feedback loop (TTFL) of the biological clock. Bmal1 forms a heterodimer with the protein product encoded by the Clock gene and activates the transcription of genes with E-box elements, such as Per 1, Per 2, Per 3, Cry 1, and Cry 2. The PER and CRY proteins then inhibit the CLOCK/BMAL1 heterodimer ac-tivity, forming a negative feedback loop [22, 23]” was added.

  1. Response to comment: As the redox and inflammatory response were measured in the current study, the introduction could benefit from adding a paragraph about the relationship between oxidative stress or inflammation and hepatic ischemia-reperfusion injury.

Response: Thank you very much for your advice. “During the ischemic phase, ATP concentration in cells decreases, resulting in the failure of the Na/K pump, cellular edoema, and an increase in cytoplasmic calcium concentra-tion, ultimately leading to cellular damage [6-8]. During the early reperfusion phase, which occurs within 2 hours of reperfusion, proinflammatory mediators (such as TNF-α, IL-6, IL-1, and arachidonic acid) and reactive oxygen species (ROS) are released by Kupffer cells. In the late reperfusion phase, which occurs between 6 and 48 hours after reperfusion, neutrophil-mediated inflammatory reactions occur. Complement factors, chemokines, and cytokines recruit neutrophils into the liver and damage cells by re-leasing ROS or proteases [5-8].” was added.

  1. Response to comment: In the section "Animal experiments ", the authors should also mention the age of animals.

Response: Thank you very much for your comments. “age: 6±1 months” was added.

  1. Response to comment: In the section "MDA detection", the authors should also mention the time of liver homogenization and the following centrifuge time for obtaining the supernatant.

Response: Thank you very much for your comments. “60Hz, 2minutes” and “centrifuged at 8000g for 10 minutes at 4 C” was added.

  1. Response to comment: In the sections "MDA or GSH detection", the authors should have used protein assay to accurately determine and then normalize the protein concentration of samples rather than using the similar weight of tissue pieces. That would be a limitation of the current study.

Response: Thank you very much for your comments. We apologize for any confusion caused by our presentation of the MDA and GSH assays. We acknowledge the reviewer’s suggestion regarding adjusting the sample concentrations to unify them according to the protein concentration before performing the subsequent assay. However, we would like to clarify that we followed the instructions provided by the kit manufacturer for calculating the results using a complex ratio of the enzyme marker assay results to the protein concentration. This calculation method is not affected by whether the initial concentration is unified or not, as stated in the kit instructions (Document 3 and Document 4). We appreciate the reviewer’s feedback and will consider their suggestion in future experiments. We will make sure to homogenize the protein concentration of each sample before performing subsequent testing. Thank you for your understanding and valuable input.

  1. Response to comment: In figure 3. the authors should show different types of tissue damage using different arrows and symbols in each representative figure. For example, cytolysis, necrosis, inflammatory cell infiltration, et.

Response: Thank you very much for your comments. As per your suggestion, I use different symbols to show the different types of tissue damage. Please see Figure 3 for details of the modifications.

  1. Response to comment: The discussion lacks a comprehensive literature search on the role of Bmal1 in controlling ischemia in other organs and tissue.

Response: Thank you very much for your comments. “The protein expression of p-Akt and Bmal1 was significantly increased in the hearts of myocardial ischemic mice. A BDNF mimetic, 7,8-dihydroxyflavone (7,8-DHF), alleviates cardiac fibrosis by restoring circadian signals via downregulating the Bmal1/Akt pathway [34]. Sun et al. found that the expression levels of Clock and Bmal1 proteins were significantly higher after 48 hours of cerebral hypoxia and ischemia [35]. Lembach et al. demonstrated that the area of the brain infarct core in Bmal1 knockout female mice was significantly smaller than that in wild-type females at 14 days after photothrombosis, confirming that Bmal1 knockout plays an active role in brain infarction [36]. Interestingly, Bmal1 knockdown in the left stellate ganglion leads to changes in the transcriptional levels of genes associated with neural activity, thereby preventing ventricular ar-rhythmias after myocardial ischemia [24]. These studies, therefore, reveal that ischemia leads to Bmal1 upregulation while Bmal1 downregulation alleviates ischemia-induced organ damage, which is consistent with our findings.” was added.

  1. Response to comment: The discussion needs detail to highlight for the reader the theme or most important pathways. For example, authors should add detailed mechanisms on how Bmal1 regulates inflammatory and oxidative signaling pathways and other involved pathways.

Response: Thank you very much for your comments. “The protein expression of p-Akt and Bmal1 was significantly increased in the hearts of myocardial ischemic mice. A BDNF mimetic, 7,8-dihydroxyflavone (7,8-DHF), alleviates cardiac fibrosis by restoring circadian signals via downregulating the Bmal1/Akt pathway [34]. Sun et al. found that the expression levels of Clock and Bmal1 proteins were significantly higher after 48 hours of cerebral hypoxia and ischemia [35]. Lembach et al. demonstrated that the area of the brain infarct core in Bmal1 knockout female mice was significantly smaller than that in wild-type females at 14 days after photothrombosis, confirming that Bmal1 knockout plays an active role in brain infarction [36]. Interestingly, Bmal1 knockdown in the left stellate ganglion leads to changes in the transcriptional levels of genes associated with neural activity, thereby preventing ventricular ar-rhythmias after myocardial ischemia [24]. These studies, therefore, reveal that ischemia leads to Bmal1 upregulation while Bmal1 downregulation alleviates ischemia-induced organ damage, which is consistent with our findings.” was added.

“Recent studies have revealed the crucial role of circadian rhythm in the production and release of collagen in fibroblasts [38]. Furthermore, a disrupted circadian rhythm has been linked to the development of various fibrotic disorders. Zhang et al. demonstrated that Bmal1 levels increased in cultured murine proximal tubular cells following TGF-β treatment and that Bmal1 deficiency protected mice against obstructive renal fibrosis [39]. Similarly, Hang et al. also found that downregulation of Bmal1 attenuated cardiac fi-brosis by inhibiting the Akt pathway [34]. Consistent with these findings, our study also demonstrated that Bmal1 downregulation significantly inhibited liver fibrosis caused by HIRI.” was added.

“In conclusion, our findings suggest that Bmal1 knockdown in the CG can amelio-rate hepatocyte injury induced by HIRI, likely by the inhibition of hepatic sympathetic nervous activity. These results suggest that targeting Bmal1 in the CG may have ther-apeutic potential for the treatment of HIRI. Further studies are needed to explore the underlying mechanisms and to develop drugs targeting Bmal1 in CG for clinical use.” was added.

  1. Response to comment: Currently, the authors only evaluated the effect of 6 hours of reperfusion. That would be interesting if authors have the data for a longer period of reperfusion to investigate the effect of their intervention in a long term. It could be pointed out in the limitation section.

Response: Thank you very much for this suggestion. “It would be interesting to assess the long-term effects of Bmal1 knockdown on longer reperfusion times.” was added.

  1. Response to comment: This study demonstrated that the downregulation of Bmal1 could ameliorate hepatic ischemia-reperfusion injury. The authors should point out the strategies to transfer their findings to the clinic in the conclusion section.

Response: Thank you very much for this suggestion. “Further studies are needed to explore the underlying mechanisms and to develop drugs targeting Bmal1 in CG for clinical use” was added.

  1. Response to comment: The manuscript would benefit from an editing service for improving readability and grammatical errors such as run-on sentences.

Response: Thank you very much for this suggestion. The article has been carefully revised by an English professional. See Document 2 for proof of modification.

Thank you very much for your review of our manuscript.

Round 2

Reviewer 1 Report (New Reviewer)

 Accept in present form

Author Response

Thank you for your patient review.

Reviewer 3 Report (New Reviewer)

Comments to the Authors 

This manuscript has been revised according to reviewer's comments. Thanks to the authors for their efforts!

Author Response

Thank you for your patient review.

This manuscript is a resubmission of an earlier submission. The following is a list of the peer review reports and author responses from that submission.

Round 1

Reviewer 1 Report

The purpose of this experiment is to study the expression of Bmal1 in CG and its regulatory role and mechanism in hepatic ischemia-reperfusion injury.

The topic is interesting and novel but some improvements are needed.

1)    The introduction is too short and more references need to be cited. For example, the article by Cannistrà M, et al. Hepatic ischemia reperfusion injury: A systematic review of literature and the role of current drugs and biomarkers. Int J Surg. 2016 Sep;33 Suppl 1:S57-70, could serve to introduce the importance to study the regulatory roles of molecules in hepatic ischemia-reperfusion injury and this can be further commented in the discussion section.

2)    Methods. Statistical analysis is poorly described as well as in the results section. Please expand.

3)    Discussion. It is too superficial in the current format. You have to compare your results with more studies of the current literature.

4)    Conclusions. You should postulate the future use of your findings in clinical contexts.